# Quinazolinones as Potential Anticancer Agents: Synthesis and Action Mechanisms

**DOI:** 10.3390/biom15020210

**Published:** 2025-02-01

**Authors:** Zhijiang Deng, Jieming Li, Pengbo Zhu, Jie Wang, Yuanfang Kong, Yulong Hu, Juntao Cai, Chunhong Dong

**Affiliations:** 1Traditional Chinese Medicine (Zhong Jing) School, Henan University of Chinese Medicine, Zhengzhou 450046, China; 2Henan Polysaccharide Research Center, Zhengzhou 450046, China; 3Henan Key Laboratory of Chinese Medicine for Polysaccharides and Drugs Research, Zhengzhou 450046, China; 4College of Pharmacy, Henan University of Chinese Medicine, Zhengzhou 450046, China

**Keywords:** quinazolinones, catalytic synthesis, anticancer, mechanisms

## Abstract

Quinazolinones, essential quinazoline derivatives, exhibit diverse biological activities with applications in pharmaceuticals and insecticides. Some derivatives have already been developed as commercial drugs. Given the rising cancer incidence, there is a critical need for new anticancer agents, and quinazolinones show promising potential in this domain. The present review focuses on novel advances in the synthesis of these important scaffolds and other medicinal aspects involving drug design, the structure–activity relationship, and action mechanisms of quinazoline and quinazolinone derivatives, to help in the development of new quinazoline and quinazolinone derivatives.

## 1. Introduction

Widdege introduced the term “quinazoline” to describe a heterocyclic compound containing a fused benzene and pyrimidine ring, also known by names such as benzimidazoline and 1,3-diazaphthalein (**1**). The properties of quinazoline derivatives are strongly influenced by the extent of conjugation and the nature and positioning of substituents on the benzene and pyrimidine rings [1]. Quinazolinones, oxidized derivatives of quinazolines, are an important class of quinazoline alkaloids. Structure–activity relationship (SAR) studies have significance of substituents at the 2, 6, and 8 positions of the quinazoline ring in determining pharmacological activity [2].

Quinazolinones are categorized based on the position of the oxygen group: 2(1H)-quinazolinones (**2**), 4(3H)-quinazolinones (**3**), and 2,4(1H,3H)-quinazoline-dione (**4**). Additionally, 4(3H)-quinazolinones are further classified by substitution patterns into 2-substituted (**5**), 3-substituted (**6**), 2,3-disubstituted (**7**), and 2,4-disubstituted (**8**) (Figure 1) [1]. Quinazolinones exhibit a broad spectrum of biological activities, including antimicrobial [2,3], antitumor [4,5,6], anti-inflammatory [7,8], anti-HIV [9,10], antimalarial [11,12], and antihypertensive [13,14] effects, making them valuable as pesticides and pharmaceuticals. Notably, some quinazolinones have been successfully commercialized as drugs (Table 1).

Cancer remains a major global health issue, affecting approximately 10 million individuals annually. The World Health Organization (WHO) reports that 14 million new cases arise each year, with projections indicating a near doubling to 30 million cases by 2040 [34]. Researchers are focusing on heterocyclic compounds including quinazolinones, as promising sources of novel anticancer agents targeting cell proliferation [35]. This review explores various synthetic approaches to quinazolinone production, including metal-catalyzed, microwave-assisted, non-metal-catalyzed, and photocatalyzed methods. Additionally, it examines the antitumor mechanisms of quinazolinone derivatives, such as microtubule polymerization, inhibition of cell cycle arrest, apoptosis induction, inhibition of tumor cell migration and invasion, suppression of angiogenesis, and targeting of epidermal growth factor receptor (EGFR) and PI3K pathways. This review aims to support research into the development of quinazolinone-based anticancer therapies.

## 2. Strategies for the Synthesis of Quinazolinones

This chapter examines catalytic synthetic methods developed for quinazolinone production, organized into metal-catalyzed, microwave-assisted, non-metal-catalyzed, and photocatalytic approaches. The aim is to inspire the development of green and efficient quinazolinone synthesis methods.

### 2.1. Metal Catalysis Reaction (Copper/Palladium/Zinc/Iron)

#### 2.1.1. Copper Catalysis Reaction

Copper offers a versatile catalytic profile, with its catalytic efficacy dependent on its oxidation state. Copper can facilitate diverse reactions, including one- and two-electron mechanisms (radical and polar) and their combinations. It readily coordinates with heteroatoms and *π*-bonds and is especially effective in activating terminal alkynes. Copper-catalyzed C–C and C–N cross-coupling reactions, such as the Ullman and Goldberg reactions, have rapidly advanced due to copper’s abundance, cost-effectiveness, and environmental sustainability compared to costly transition metals [36]. This has spurred a surge in research into copper-catalyzed organic reactions. Deng et al. reported an efficient, eco-friendly, copper-catalyzed tandem reaction of 2-aminobenzamide with tertiary amine using a one-pot method involving cyclization and coupling to produce various quinazolinone derivatives (Figure 2). Conducted under aerobic conditions, this reaction achieved yields of up to 91% [37]. Similarly, Bao et al. developed a method for synthesizing 2-substituted 4(3H)-quinazolinones from readily available starting materials (Figure 3). This process involves copper Lewis acid-catalyzed nucleophilic addition of 2-hexanediolamides to nitriles, followed by an intramolecular SNAr reaction using *^t^*BuOK, noted for its simplicity and accessibility of raw materials [38]. Additionally, Upadhyaya et al. synthesized 2-substituted quinazolinones from 2-halo benzamide using TMSN3 as a nitrogen source, without ligands or bases, in a single reactor via a series of coupling steps, including oxidative addition, reductive amination, phenyl C(sp3)-H bond oxy-nitridation, intramolecular cyclization, and oxidative dehydrogenation [39].

Jurriën W. Collet et al. developed a method for synthesizing quinazolinones using anisole as a green solvent, eliminating the need for drying conditions and inert gases. This reaction employs Cu(II) acetate as a catalyst and a mild base, to facilitate cross-coupling between 2-isocyanatophenone and amines, leading to a cyclo-condensation reaction (Figure 4) [40]. Readily available 2-aryl indoles can also react with amines or ammonia through successive Baeyer–Villiger oxidative expansions and dehydrative condensation reactions under O_2_ conditions to form quinazolinones (Figure 5). This method is notable for its straightforward process, mild reaction conditions, and environmentally benign characteristics [41].

Liu et al. synthesized 11H-pyrido[2,1-b]quinazolin-11-ones using Cu(OAc)_2_·H_2_O as the catalyst, with substituted isatins and 2-bromopyridine derivatives as starting materials (Figure 6) [42]. This reaction involves C–N and C–C bond cleavage along with the formation of two C–N bonds, offering flexibility for further functionalization across various substrates. The use of the inexpensive Cu(OAc)_2_·H_2_O catalyst yields good to excellent results. Bao et al. presented a simple method for synthesizing N-substituted quinazolinones from anthranilamides using DCP as the methyl source (Figure 7). This reaction involves a tandem process of N-methylation, C(sp^3^)-H amination, and oxidation, facilitated by a copper catalyst [43].

#### 2.1.2. Palladium-Catalyzed Reaction

Palladium-catalyzed coupling reactions are widely utilized in both industrial and laboratory synthesis, offering an efficient approach for constructing C–C bonds and C–heteroatom. Jiang et al. developed a palladium-catalyzed method to synthesize quinazoline-4(3H)-ones from 2-aminobenzamides and aryl halides (Figure 8). The optimal reaction conditions included three equivalents of aryl halides, three equivalents of tert-butyl isocyanide, 5 mol%PdCl_2_, 0.1 equivalent of DPPP, two equivalents of CaCl_2_, and four equivalents of NaOtBu in toluene at 145 °C for 8 h, achieving moderate to excellent results [44]. Qian et al. synthesized 2,3-disubstituted quinazolinones through a palladium-catalyzed, three-component oxidative coupling of anthracene amide with isocyanate and aryl boronic acid (Figure 9) [45]. Additionally, Qiu et al. developed a high-yield, one-pot, palladium-catalyzed, three-component reaction of bis-(2-iodoaromatic)-carbodiimide, isocyanide, and an amine. This approach combines nucleophilic attack, isocyanide insertion, and C–N coupling to generate quinazolino[3,2-a] quinazolines and related compounds in high yields (Figure 10) [46].

Further advancements include an efficient palladium-catalyzed cyclization method that synthesizes fused quinazolinone derivatives from 3-arylquinazolinones through mono- and double-alkyne insertions in a one-pot process. This reaction, featuring C–X cleavage, alkyne insertion, 1,4-palladium migration, and C–H annulation, yielded products in 43–80% yields (Figure 11) [47]. Dabiri et al. synthesized novel substituted hydroxyisoindolo[1,2-b]quinazolinone compounds through a palladium-catalyzed cross dehydrogenative coupling (CDC) reaction between aryl-quinazolinones and aldehydes, achieving moderate to high yields [48]. In a variation, they replaced benzaldehyde with toluene and used excess tert-butyl hydroperoxide (TBHP) as the oxidant, which significantly improved the yields (Figure 12).

#### 2.1.3. Zinc-Reduced Synthesis

Zinc was the first metal identified to participate in water-phase Barbier reactions, and it effectively catalyzes reactions such as allylation and benzylation of carbonyl compounds, along with specific alkylation. Additionally, zinc catalyzes carbon–nitrogen double-bond Barbier reactions, including the allylation of imines and α-amino-aldehydes, showing enhanced activity in aqueous stabilization [49]. In their study, L-M. Wang et al. utilized a novel catalyst, zinc(II) perfluorooctanoate [Zn(PFO)_2_], in a one-pot, three-component cyclo-condensation reaction, achieving high yields of quinazolinone derivatives through atom-efficient synthesis in an aqueous micellar medium (Figure 13) [50].

Manna et al. identified a radical-mediated pathway for quinazolinones quinazolinone synthesis using redox-active amide ligand-loaded zinc compounds. This method facilitates the deamination of *o*-amino amide/ester derivatives, enabling coupling with the amino group of the nitrile (Figure 14) [51]. Das et al. reported that zinc catalyzes the cyclization of ureido benzoates to produce quinazoline diketones under mild conditions (Figure 15). This approach allows the efficient synthesis of quinazolinone derivatives from various acyclic ureido benzoates containing diverse electron-donating and electron-withdrawing groups. Using low-toxicity, accessible zinc salts such as Lewis acids, their method offers a cost-effective and time-efficient alternative to traditional synthetic strategies [52].

#### 2.1.4. Iron-Catalyzed Reaction

Iron compounds valued for their abundance and low cost are widely used in organic synthesis and have critical roles in biological systems. In lower oxidation states, iron can act as a nucleophilic reagent, catalyzing a variety of reactions. Iron’s potential as a catalyst in drug, material, and functional molecules is extensive, with the field rapidly progressing [53]. Ghouse et al. developed an Fe(II)-catalyzed cascade reaction for cyanoalkyl sulfonylation and cyclization, enabling the synthesis of functionalized sulfonated quino-quinazolidinone alkyl cyanide (Figure 16). This three-component radical transformation achieves high chemo- and regioselectivity without external oxidizing or reducing agents [54].

Wu et al. reported an Fe(III)-mediated cascade/decarbonylation cyclization reaction for synthesizing 2-(trifluoromethyl)quinazoline-4(3H)-ones using isatins and trifluoroacetic acid chlorides as starting materials (Figure 17). The reaction forms a biologically significant quinazoline-4(3H)-one derivative through an efficient pathway involving tricyclic amphiphilic intermediates that undergo intramolecular nucleophilic attack and subsequently release 2-(trifluoromethyl)quinazoline-4(3H)-one alongside carbon monoxide [55]. Serva et al. presented an efficient synthesis of 2-substituted quinazolin-4(3H)-ones by reacting isatoic anhydride with various amidoxime derivatives catalyzed by iron(III) chloride [56]. Malviya et al. established a non-stereoselective approach to synthesizing schizocommunin derivatives through an iron-catalyzed oxidative dehydrogenation coupling reaction between two different C(sp^3^)-H bonds (Figure 18), producing a broad range of substituted indol-2-ones and quinazolines using air as an eco-friendly oxidant [57].

Ding et al. introduced an efficient iron-catalyzed cross-dehydro-coupling of [4+2] cyclized secondary/tertiary anilines with quinazolinones to produce quinoline-spiro-quinazolinones [58]. Using FeCl_3_ as the catalyst and H_2_O_2_/O_2_ as the oxidant in ethanol at room temperature, this method accommodates a wide range of functional groups. N-methylanilines were incorporated into tetrahydroquinolines, saving methyl and methylene sources for cycloaddition reactions (Figure 19). This environmentally friendly process shows significant promise for applications in the agrochemical and pharmaceutical industries.

The Ru(II)-catalyzed C–H activation of quinazolinones combined with an olefinic difunctional merger is a promising approach for the redox-neutral synthesis of dihydro-isoquinazolidinone derivatives [59]. This intermolecular reaction proceeds rapidly, yielding products in high quantities without the need for a stoichiometric metal oxidizer. In a similar vein, silver-catalyzed hydroalkoxylation of C2-alkynyl quinazolidinones enables the selective formation of octa-membered N, O-heterocycles via 8-endo cyclization, producing the target compounds with high efficiency [60]. Furthermore, gold-catalyzed selective hydrogenation of alkynyl quinazolinone ether pyrroles, followed by rearrangement and cyclization, successfully synthesizes 1,2- and 2,3-fused quinazolinones [61]. Additionally, Co(III)-catalyzed cyclization of 2-aryl quinazolinones with alkynes demonstrates compatibility with various functional groups, including both electron-donating and electron-withdrawing groups, as well as halides, and exhibits high regioselectivity with asymmetric alkyne substrates [62].

While the metal-catalyzed synthesis of quinazolinones has achieved remarkable results, there are still some areas that need to be improved and optimized. The efficiency and sustainability of the metal-catalyzed synthesis of quinazolinones can be further improved through in-depth study of the reaction mechanism, optimization of the reaction conditions, development of catalysts that can be easily recycled and reused, and promotion of the development of green chemistry.

### 2.2. Microwave-Assisted Synthesis

Microwave technology, situated in the electromagnetic spectrum between infrared and radio waves, is gaining attention for its ability to rapidly and uniformly heat chemical reaction systems. This process enhances efficiency, reduces energy consumption, and is more environmentally friendly [63]. Fozooni et al. synthesized dihydro-quinazoline derivatives containing oxazolone rings under both microwave irradiation and reflux conditions [64]. They catalyzed a one-pot, three-component reaction using acetic acid, isophthalic anhydride, an aromatic aldehyde, and 4-aminomartinine to produce 2,3-dihydroquinazoline-4(1H)-ones. These derivatives were then combined with aromatic aldehydes, acetic anhydride, and sodium acetate to yield the oxazolone derivative (Figure 20). Notably, microwave-assisted conditions enhanced the compound yield compared to conventional methods.

Benzyl alcohol and aminobenzamide derivatives were used to synthesize quinazolinones under microwave conditions (Figure 21). A novel water-assisted method was developed that incorporated sodium chloride as a salting agent and tert-butyl peroxide as an oxidant, demonstrating improved efficiency over traditional methods [65]. The optimal conversion to the desired product occurred at 400 W and 80 °C. J-L Wu et al. successfully synthesized keto-alkyl-substituted polycyclic quinazolinone derivatives via microwave irradiation, employing a radical cascade alkylation/cyclization of inactivated olefins with ketone as the alkylating agent (Figure 22) [66]. This method is transition-metal-free, compatible with a wide range of functional groups, and straightforward in execution.

Murhta et al. synthesized diverse, novel thiazole[2,3-b]quinazolinone derivatives using an acid-mediated, one-pot domino reaction under microwave irradiation with 2-amino-substituted thiazoles, substituted benzaldehydes, and cyclic diketones (Figure 23) [67]. The process involved acid-catalyzed condensation of dienones with aromatic aldehydes, forming Michael adduct intermediates, which then reacted with 2-amino-4-arylthiazoles to create aminothiazole derivatives. These underwent deprotonation, followed by intramolecular cyclization and keto-enol tautomerization to yield thiazole[2,3-b]quinazolinones.

Microwave heating enables uniform heating across the reaction system, accelerating reaction rates and boosting yields. Polar solvents, such as water or alcohol, are particularly effective in microwave catalysis as they absorb microwave energy efficiently, which activates the reactants and increases the reaction speed. The alignment of polar molecules under microwave radiation promotes effective collisions, thus accelerating the reaction. Compared to metal-catalyzed reactions, microwave-assisted synthesis offers a more efficient, eco-friendly alternative, eliminating the need for high temperatures and extended reaction times. However, there are also limitations, including high equipment costs, difficulties in controlling reaction conditions, limited applicability, and safety concerns. To further enhance the reliability and application value of microwave-assisted synthesis methods, measures such as optimizing the design of microwave reactors, developing novel microwave-absorbing materials, strengthening control over reaction conditions, and expanding the scope of application can be taken.

### 2.3. Metal-Free Catalytic Reaction

The application of metal-free catalysis in organic synthesis is gaining traction due to its unique advantages, establishing it as a highly valued catalytic approach [68]. Metal-free catalysts typically offer excellent recoverability, as they are easier to separate and recycle compared to metal catalysts, reducing both catalyst costs and environmental impact. This approach aligns well with green chemistry and principles, eliminating the need for complex reaction conditions and post-processing steps, and simplifying laboratory operations.

Wang et al. developed an efficient, atom-economical method for synthesizing quinazolinones via an aerobic oxidative cascade ring reaction of transition-metal-free aminonitriles and alcohols [69]. When utilizing atmospheric oxygen as an oxidant under mild conditions, the reaction produces only water as a by-product (Figure 24). With its broad substrate scope and absence of transition metal residues, this method holds promise for pharmaceutical applications. Luo et al. employed tert-butyl hydroperoxide as an oxidant to catalyze the reaction between quinazoline-3-oxides and primary amines, synthesizing quinazolin-4(3H)-ones in high efficiency and adaptability across a wide range of primary amines (Figure 25) [70].

Zhang et al. introduced a novel metal-free catalytic method for the intramolecular oxidative C-H amination of (E)-3-(aryl-amino)-2-styrylquinazoline-4(3H)-ones, enabling the synthesis of 1,2-diarylpyrazolo[5,1-b]quinazoline-9(1H)-ones with high yields [71]. This method accommodates various functional groups and provides a new route for synthesizing 2,3-fused quinazolinones (Figure 26). Additionally, the approach is scalable to gram quantities, making it feasible for larger-scale applications. The reaction between methyl 2-isothiocyanatabenzoate and 1-azido-3-(4-substituted phenyl)propan-2-ones was conducted under heating with triphenylphosphine in a dioxane solvent, yielding tricyclic dihydro-5H-oxazolo[2,3-b]quinazolin-5-ones (Figure 27) [72].

Liu et al. further advanced metal-free synthesis with a one-pot method for producing quinazolino[3,4-*α*]quinazolin-13-ones by combining *o*-(methoxycarbonyl)benzene-diazonium salts, nitriles, and 2-cyanoanilines (Figure 28). This approach synthesized diverse polycyclic scaffolds through amination, cyclization and amidation steps, forming nitro-carbonitrile intermediates from diazonium salts and nitrile-functionalized intermediates [73]. Noteworthy for its mild conditions, broad substrate compatibility, and simplicity, this method efficiently forms four consecutive C-N bonds, underscoring its potential for further synthetic applications.

Li et al. reported an efficient, selective synthesis of quinazolinones via C–C bond cleavage through a phosphoric acid-catalyzed cyclo-condensation of *β*-ketoester and *o*-aminobenzamide [74]. This metal- and oxidant-free reaction provides both 2-alkyl- and 2-aryl-substituted quinazolinones with high yields and excellent selectivity, showing broad substrate compatibility (Figure 29). This approach is adaptable to the synthesis of other N-heterocycles, benzimidazole and benzothiazole. A novel synthesis method for spiroiso indolone dihydro-quinazolinones was developed using KHMDS as a base with 2-aminobenzamide and 2-cyanoethyl benzoate as substrates (Figure 30). Mechanistic studies indicate that KHMDS deprotonates the N-H bond, facilitating nucleophilic attack by nitrogen on cyano-benzoate. This forms an imine intermediate, which cyclizes to yield spiroiso indolone dihydro-quinazolinones [75].

Jayaram et al. demonstrated a direct, metal-free synthesis of acylated and alkylated quinazolinone derivatives using 2-amino benzamides with I_2_/DMSO and epoxides via ring-opening reactions (Figure 31) [76]. Here, I_2_ catalyzes an oxidative coupling between 2-aminoformamide and aryl methyl ketones to form 2-aryl quinazolin-4(3H)-ones. This method features high functional group tolerance, substrate selectivity, and significant yield improvements with continuous flow technology, which reduces reaction times.

In catalyst-free conditions with water as a solvent, α-keto acids react with 2-amino benzamides to form quinazolinones (Figure 32). Similarly, 2-amino thiophenol, benzene-1,2-diamine, and 2-aminophenol react to yield benzothiazoles, quinoxalinones, and benzo-xazinones, respectively. Purification techniques, such as filtration, ethanol washing, or crystallization, finalize the products [77].

These metal-free reactions, which employ small organic molecules like amino acids, olefins, and alcohols, rely on hydrogen bonding and electron transfer to produce intermediates, particularly imines from amines and aldehydes. Such reactions offer milder conditions, making them suitable for thermally sensitive substrates and appealing for green chemistry applications. Although metal-free catalysis eliminates the need for precious metal catalysts, boasts simpler operational steps, reduces costs and the complexity of synthesis processes, and minimizes the risk of environmental pollution, the stability and regenerability of organic catalysts may not compare to metal catalysts, necessitating further improvement and optimization.

### 2.4. Photocatalytic Reaction

Photo-redox chemistry relies on photoexcited catalysts that facilitate single-electron and energy transfer reactions with organic molecules. This energy transfer serves as a key decay pathway for photoexcited states, operating as a catalytic mechanism in photo-redox processes. Xu et al. developed a visible light-catalyzed deamidation of aniline, enabling its incorporation into 1,2,3-benzotriazoles (Figure 33). This reaction involves energy transfer from photoexcited Ir(ppy)3, which excites 1,2,3-benzotriazinone, initiating a denitrogenation rearrangement and producing quinazolinone derivatives [78]. This approach is notable for its mild conditions, simplicity, and versatility, allowing for the synthesis of diverse quinazolinone derivatives. Wang et al. introduced an eco-friendly synthesis of quinazolinones by condensing o-aminobenzamide and aldehydes under visible light using fluorescein as a photocatalyst [79]. This reaction is scalable to gram quantities and holds promise for industrial applications.

Anandhan et al. reported a visible-light-driven oxidative cleavage of the C–N bond in N,N-di-phenylaniline, forming secondary amides. They also synthesized quinazolinones from 2-(di-benzylamine)benzamide utilizing (NH_4_)_2_S_2_O_8_ as an additive and Rose Bengal for the regioselective oxidation of N-benzyl tertiary amines, achieving secondary amides via C–N bond cleavage (Figure 34) [80]. Sun et al. presented a catalyst-free photochemical method for synthesizing polyfluoroalkylated quinazolinone derivatives (Figure 35). This reaction, initiated by a 10 W LED, yields trifle-oro methylated quinazolinones with moderate to high efficiency, up to 83% [81]. Its compatibility with a wide range of functional groups and gentle conditions makes it highly advantageous, utilizing cost-effective, readily available reagents.

Additionally, under visible light, 3-(2-(ethyl)phenyl)quinazolinones and di-aryl-phosphine oxides were synthesized through a phosphorylation/cyclization reaction using 4CzIPN as a photocatalyst (Figure 36) [82]. Enhanced reaction efficiency was achieved under continuous flow conditions, offering benefits such as metal-free catalysis, broad substrate compatibility, ambient temperature operation, scalability, and the potential use of sunlight as an irradiation source. Gao et al. developed a phosphine radical-triggered cascade addition/cyclization reaction of non-activated olefins, employing TBPB as an oxidant. This mild method produced biologically significant, phosphine-containing quinazolinones via highly reactive phosphine oxides [83].

In photocatalytic reactions, light absorption by catalysts is crucial; the choice of catalyst impacts reaction efficiency based on absorption capacity. Fluorescein, for example, demonstrates high visible light absorption, making it effective for quinazolinone synthesis. Upon excitation, electrons transfer across on the catalyst surface, producing reactive species. These excited catalysts interact quickly with other molecules, generating highly reactive free radicals. Catalysts such as TiO_2_ and fluorescein, which are highly active under visible light, outperform metallic catalysts by reducing energy consumption and minimizing side reactions. As eco-friendly alternatives, photocatalytic reactions avoid harsh acids and oxidizing agents, reducing environmental pollution. However, there are also limitations, including limited photocatalyst options, specific light source requirements, sensitivity to reaction conditions, substrate applicability, as well as challenges in product separation and purification.

## 3. The Antitumor Mechanism of Quinazolinones

Quinazolinones exhibit a wide range of pharmacological activities, including notable anticancer effects against cancers such as breast, lung, and pancreatic malignancies. This promising activity has driven ongoing research to develop quinazolinone-based anticancer agents (Table 2). This chapter reviews the current studies on the antitumor mechanisms of quinazolinones, aiming to inspire advancements in anticancer drug development.

### 3.1. Tubulin Polymerization Inhibitor

Quinazolinone can inhibit tubulin polymerization, a process critical for cell division, by preventing the assembly of tubulin into functional microtubules. Given that uncontrolled cell division is a hallmark of cancer, this property of quinazolinones has therapeutic potential for anticancer drug design [84]. Raffa et al. synthesized a novel quinazolin-4(3H)-one derivative, compound (**101**), by reacting 6-chloro-2-methyl-3-(heteroaryl)-4(3H)-quinazolinone with benzaldehyde in glacial acetic acid [85] (Figure 37). At a concentration of 1 µg/mL, compound (**101**) inhibited proliferation in L1210 and K562 leukemia cell lines by more than 50%, with an IC_50_ value of 5.8 µM. For comparison, colchicine displayed an IC_50_ of 3.2 µM. Compound (**101**) also inhibited the growth of MCF-7 and Burkitt lymphoma CA46 cells, with IC_50_ values of 0.34 µM and 1.0 µM, respectively. Additionally, (**101**) induced a G2/M phase arrest in Burkitt lymphoma cells at 10 µM, resulting in a 20% mitotic index, and disrupted microtubule structures in MCF-7 cells at 3.4 µM.

Yang et al. investigated the signaling pathways involved in microtubule interactions and apoptosis using 6-pyrrolidinyl-2-(2-hydroxyphenyl)-4-quinazolinone (compound **102**) in U937 xenograft models (Figure 37). Compound (**102**) showed strong binding to *α*- and *β*-tubulin proteins, inhibiting microtubule polymerization in vitro and in vivo. This interference led to disordered microtubule organization and mitotic arrest via activation of the CDK1/cyclin B complex [86]. Potter et al. developed quinazolinone derivatives as part of a series of 2-methoxyestradiol-based microtubule disruptors. In vitro testing on DU-145 prostate and MDAMB-231 breast cancer cell lines revealed that these compounds had more potent antiproliferative effects than combretastatin. Specifically, the 2′-methoxy derivatives formed hydrogen bonds with tubulin receptors and disrupted microtubules, displaying notable anticancer activity [87]. Compounds (**103**) and (**104**), in particular, effectively inhibited microtubule assembly with an IC_50_ that was 2–3 times lower than that of CA-4, indicating their potential to destabilize microtubule dynamics and arrest the cell cycle (Figure 37).

In a recent study, Shi et al. designed and synthesized a series of 2-substituted 2,3-dihydroquinazolin-4(1H)-one derivatives (Figure 37) [88]. Mechanistic studies revealed that compound (**105**) significantly inhibited microtubule protein polymerization in vitro, disrupted the cellular microtubule structure, induced G2/M-phase cell cycle arrest, and triggered apoptosis through the upregulation of cleaved PARP-1 and caspase-3. Molecular docking analyses confirmed that compound (**105**) effectively occupied the binding site of microtubule proteins. These findings suggest that 2-dihydroquinazolin-4(1H)-one derivatives with benzene, biphenyl, naphthyl, or indolyl groups at the C2 position could represent a novel class of antitumor agents targeting microtubule protein polymerization.

In a quinazolinone library of 59 compounds, structure (**106**) was identified through virtual ligand screening as an inhibitor of cell-cycle-dependent kinase 4(Cdk4) and microtubules [89]. Compound (**106**) demonstrated Cdk4 inhibition with an IC_50_ of 0.47 µM, and microtubule polymerization inhibition with an IC_50_ of 0.6 µM (Figure 37). Cancer cell cycle analyses confirmed that this dual inhibition blocks cells in the G0/G1 and M phases. These results underscore the potential of virtual screening in designing novel inhibitors that effectively target key phases of the cell division cycle.

Further analysis of the conformational relationships in quinazolinones with enhanced microtubule polymerization inhibition revealed that aryl rings substituted with halogens, nitro groups, or methoxy groups improve microtubule binding. Substitution at the 2-position with alkyl or heteroaryl groups can significantly enhance inhibition. Polar groups at the 6- or 8-position increase water solubility and bioavailability while maintaining potent inhibitory activity. Introducing aryl ring structures and hydrophobic substituents further strengthens the inhibitory effect, and compounds with specific chiral features exhibit even greater activity.

### 3.2. Initiation of Cell Cycle Arrest

Research increasingly supports the ability of quinazolinones to induce cell cycle arrest, often leading to apoptosis (programmed cell death) and reduced tumor cell viability.

Le et al. designed and synthesized a series of novel 3-methylquinazolinone derivatives to evaluate their antitumor activity against the wild-type epidermal growth factor receptor tyrosine kinase (EGFRwt-TK) [90]. Testing these compounds on human cancer cell lines A549, PC-3, and SMMC-7721 revealed that compound (**107**) induced late apoptosis and caused G2/M phase cell cycle arrest in A549 cells at higher concentrations (Figure 38). Additionally, it inhibited EGFRwt-TK with an IC_50_ of 10 nM. In another study, Ali et al. synthesized a series of 2-thioquinazolin-4(3H)-one conjugates and assessed their efficacy against multiple cancer cell lines [68]. The benzimidazole-quinazolinone derivative (**108**) exhibited notable anticancer activity and targeted RAF kinase. Compound (**108**) effectively induced G2/M phase cell cycle arrest and triggered apoptosis in A-375 melanoma cells, implicating it as a promising therapeutic agent. It also upregulated RecQ deconjugating enzymes, known to play roles in cancer cell survival, highlighting a potential therapeutic target.

Haggag et al. designed quinazolinone derivatives evaluated by the National Cancer Institute (NCI) for cytotoxicity across 60 cancer cell lines and for their inhibitory effect on BLM-deconjugating enzymes. Compound (**109**) demonstrated moderate activity, inhibiting WRN and RECQ1 conjugates and interacting with the ATP binding site of RecQ conjugate enzymes (Figure 38) [91]. In HCT-116 and MDAMB-231 cell lines, compound (**109**) induced G2/M phase arrest and apoptosis, indicating its high anticancer potential, favorable safety profile, and selective inhibition of pan-RecQ-deconjugating enzymes.

Liu et al. investigated the effects of a synthetic quinazolinone analog, 2-(naphthalen-1-yl)-6-pyrrolidino-4-quinazolinone (compound **110**), on glioma cells (Figure 38) [92]. Treatment with compound (**110**) induced cell death associated with a multinuclear phenotype and multipolar spindles, arresting the cell cycle in the G2/M phase and increasing polyploidy. Western blot analysis showed elevated levels of cell cycle proteins B1, Cdk1, and pY15 after treatment, suggesting that compound **110**’s antitumor activity in glioma may be due to its interference with the G2/M checkpoint, presenting a promising new direction for glioma therapies.

### 3.3. Induction of Apoptosis

Apoptosis, a critical biological process governing cell survival and death, is mediated by two primary pathways: exogenous (death receptor-mediated) and endogenous (mitochondrial and pro-apoptotic factor-mediated) [93].

Xie et al. synthesized a series of 4-[(*α*-l-rhamnosyloxy)benzyl]isothiocyanate (MITC) quinazolinone derivatives and investigated the anticancer properties of the most active compounds [94]. Derivative (**111**) inhibited the growth of several types of cells and induced apoptosis in U251 cells, as indicated by changes in caspase-3 and Bax: Bcl-2 changes (Figure 39). The Bcl-2 ratio of U251 cells showed a notable decline in the G1 phase and a rise in the S and G2 phases, along with reduced levels of cell cycle proteins. These findings suggest that derivative (**111**) holds the potential for glioma prevention and treatment.

The 6,7-disubstituted-2-(3-fluorophenyl)quinazolinone derivative (**112**) significantly inhibited oral squamous cell carcinoma (OSCC) cell viability (Figure 39) [95]. This compound induced G2/M-phase cell cycle arrest, upregulated cell cycle protein B, enhanced Ser10 phosphorylation of histone H3, and facilitated PARP cleavage, all indicative of mitotic arrest followed by apoptosis. Furthermore, combining derivative (**112**) with 5-fluorouracil (5-FU) resulted in a synergistic cytotoxic effect on OSCC cells.

In their study, Lu et al. explored the mechanism of action for quinazolinone derivative (**113**) in human oral cancer CAL 27 cells, identifying the roles of intrinsic molecules during treatment (Figure 39) [96]. Exposure to compound (**113**) activated Cdk1, which regulated Bcl-2-mediated mitotic arrest and apoptosis through Ser70 phosphorylation, leading to cell death. This pathway also triggered intracellular Ca^2+^ release and activated markers of endoplasmic reticulum stress. In vivo application of compound (**113**) led to a marked G2/M-phase blockade and significant tumor suppression in nude mice bearing CAL 27 tumors.

Zahedifard et al. discovered that dihydroquinazolin-4(1H)-one derivatives (**114**) and (**115**) significantly inhibited effects on MCF-7 cell viability (Figure 39) [97]. Using the Cellomics high-content screening (HCS) method, they demonstrated that these compounds facilitate the translocation of cytochrome c from the mitochondria to the cytoplasm, triggering the activation of caspase-9 and subsequently caspase-3/7. Additionally, caspase-8 activity was markedly increased in MCF-7 cells treated with these compounds, leading to the inhibition of NF-κB activation. These results suggest that compounds (**114**) and (**115**) may induce apoptosis via both exogenous and endogenous pathways.

Kumar et al. investigated the interdependence of autophagy and apoptosis in human leukemia MOLT-4 cells when treated with a novel quinazolin-4(1H)-one derivative (**116**) (Figure 39) [98]. Their findings showed that compound (**116**) induced a cytochrome c-mediated apoptosis and autophagy mechanism. Compound **116**’s autophagic potential was verified through acridine orange staining, LC3 immunofluorescence, and Western blot analyses, revealing that cytochrome c acts as a negative feedback regulator of autophagy. The study highlighted the importance of quinazolinone derivatives in developing new anticancer drugs.

Quinazolinone compounds have also been shown to block cell cycle progression, especially in the G2/M phase, by inhibiting RecQ dissociation and upregulating cell cycle regulatory proteins such as B1, Cdk1, and pY15, collectively hindering tumor cell proliferation. Additionally, these compounds may promote apoptosis in tumor cells by enhancing the expression of pro-apoptotic proteins, such as caspases, while simultaneously inhibiting anti-apoptotic proteins like Bcl-2 and disrupting cell cycle regulatory proteins. The capacity of quinazolinone compounds to induce apoptosis in tumor cells through endogenous or exogenous pathways remains a key focus in assessing their antitumor potential.

### 3.4. Acting in the TME

The tumor microenvironment (TME) comprises diverse cell types, including immune cells, cancer-associated fibroblasts (CAFs), endothelial cells (ECs), pericytes, and other additional tissue-resident cells [99]. CAFs are central to the TME, aiding tumor growth and survival by secreting growth factors, cytokines, and stromal components. Within the TME, CAFs can become activated through exposure to inflammatory mediators and alterations in the extracellular matrix (ECM). Soluble activators like TGF-β, IL-1, IL-6, and TNF-α drive chronic inflammation and are pivotal in activating CAFs, which support tumor development and progression [100]. The stiffness of the ECM further promotes gene reactivation in CAFs, initiating pro-fibrotic responses, ECM protein production, angiogenesis, and cancer cell invasion [101].

Dahabiyeh et al. evaluated seven synthetic 2,3-dihydroquinazoline-4(1H)-one analogs for their efficacy against PC3 and DU145 cancer cells [102]. Through MTT assays, scratch healing, adhesion and invasion assays, and LC-MS analyses, they identified that compound (**117**) had notable inhibitory effects on prostate cancer cell adhesion and invasion, achieving an IC_50_ below 15 µM (Figure 40). Metabolic profiling of compound (**117**) revealed impacts on energy production, redox status, amino acid and polyamine metabolism, and choline-phospholipid homeostasis processes essential for cancer cell growth and proliferation.

Abdallah et al. designed quinazolinone derivatives to explore their immunomodulatory potential, incorporating glutarimide fragments to mimic thalidomide’s effects [103]. These compounds were tested on breast (MCF-7), colorectal, liver (HepG-2), and prostate (PC3) cancer cells. Compound (**118**), in particular, showed a lower IC_50_ than thalidomide and significantly reduced TNF-α and IL-6 levels while increasing caspase-3 levels sixfold (Figure 40). The anti-inflammatory effects of quinazolinone derivative (**118**) were attributed to its suppression of pro-inflammatory mediators such as nitric oxide synthase-II (NOS-II) and TNF-*α*.

Askar’s team synthesized a series of quinazolinone benzene sulfonamide derivatives, subjecting them to MTT assays [104]. Compounds (**119**) and (**120**) showed growth-inhibitory effects on HepG-2 and MCF-7 cells with IC_50_ values of 6.65 µM and 8.27 µM, respectively (Figure 40). Both compounds also exhibited immune-stimulating effects on CD4+ and CD8+ T lymphocytes, demonstrated by increased spleen and thymus weights, suggesting their potential as promising candidates for further anticancer drug development.

### 3.5. Angiogenesis Inhibitor

Angiogenesis plays a vital role in cancer progression by providing tumor cells with oxygen and nutrients and facilitating metastasis [105]. This process is regulated by a delicate balance of pro-angiogenic factors, like Vascular Endothelial Growth Factor (VEGF) and Basic Fibroblast Growth Factor (BFGF), and anti-angiogenic factors, such as Angiogenesis Inhibitor-2 (Ang-2). Disruption of this balance can promote tumor growth and metastasis [106].

Ghorab et al. synthesized the compound 4-(2-(2-hydrazinyl-2-oxoethylthio)-4-oxobenzo[g]quinazolin-3(4H)-yl) benzene sulfonamide (**121**) and assessed its cytotoxicity effects on MCF-7 breast cancer cells and its inhibitory action on VEGFR-2 (Figure 41) [107]. Compound (**121**) showed significant VEGFR-2 inhibition with an IC_50_ of 0.64 µM, alongside increased caspase-3 activity, elevated Bax levels, reduced BCl2 levels, and induced G2/M phase cell cycle arrest. Zahran et al. further developed a sulfon chloropyrazine-containing quinazolinone (**122**) and evaluated it against 60 cancer cell lines [108]. Compound (**122**) demonstrated potent VEGFR-2 inhibition with an IC_50_ of 66 ± 0.002 nM, effectively inhibited cell migration, induced S-phase cell cycle arrest, and promoted apoptosis across various stages, as indicated by the Annexin V-FITC assay. In UO-31 cells, compound (**122**) treatment increased caspase-3 activity and modulated Bax and Bcl-2 expression.

In another study, Mahdy et al. synthesized quinazoline-4(3H)-one derivatives as VEGFR-2 inhibitors, testing their efficacy against HepG2, HCT-116, and MCF-7 cancer cells [109]. Compound (**123**) showed IC_50_ values of 3.97 µM, 4.58 µM, and 4.83 µM in these cell lines, respectively, and a VEGFR-2 inhibition IC_50_ of 2.5 µM, comparable to sorafenib (IC_50_ = 2.4 µM) (Figure 41). Compound (**123**) also caused G2/M-phase arrest in HepG-2 cells, and molecular docking revealed a favorable binding affinity to VEGFR-2’s active site, with a binding free energy (ΔG) of −59.90 Kcal/mol, comparable to sorafenib’s ΔG = −52.20 Kcal/mol. The identification of compound (**123**) represents a promising advancement in this area of research.

Pathak et al. synthesized a series of quinazolinone-substituted 1,3,5-triazine derivatives with notable VEGFR-2 inhibitory activity [110]. Among them, compound (**124**) exhibited significant antitumor effects against MCF-7, HL-60, and HeLa cell lines, with IC_50_ values of 8.69, 8.40, and 6.65 µM, respectively (Figure 41). It also demonstrated anti-angiogenic efficacy comparable to vandetanib. Molecular docking analyses highlighted that compound (**124**) formed hydrogen bonds with VEGFR-2 residues, yielding lower binding energies than vandetanib, underscoring its potential as a promising anti-angiogenic agent.

Research indicates that quinazolinone compounds can play a crucial role in inhibiting angiogenesis, thereby impacting tumor growth and metastasis. Quinazolinone derivatives have shown efficacy in reducing the expression of BFGF, thereby diminishing angiogenic capacity. By inhibiting VEGF production, these compounds disrupt the angiogenic signaling pathway, ultimately affecting the tumor’s blood supply. Additionally, quinazolinone derivatives can significantly lower levels of the pro-inflammatory cytokine IL-8, further deducing angiogenesis. Through the downregulation of pro-angiogenic factors like BFGF, VEGF, and IL-8, quinazolinone derivatives effectively slow tumor growth. This anti-angiogenic effect not only limits the development of primary tumors but also holds promise in preventing and treating metastatic tumors. When used in combination therapy, the anti-angiogenic properties of quinazolinone derivatives may enhance the efficacy of other anticancer treatments. By reducing the tumor’s blood supply, they can improve the effects of chemotherapy and radiotherapy, while potentially lowering the risk of drug resistance. Future studies will aim to deepen the understanding of these mechanisms, further advancing quinazolinone derivatives in cancer therapy.

### 3.6. EGFR Inhibitor

The EGFR is a critical transmembrane receptor weighing 170 kDa that regulates essential cellular processes such as proliferation, survival, and migration [111]. EGFR overexpression, due to gene amplification, mutations, or increased ligand expression, contributes to excessive signaling in cancer cells, making EGFR a significant target in cancer treatment [112]. Anti-EGFR monoclonal antibodies (MoAbs) and small-molecule EGFR tyrosine kinase inhibitors (TKIs) are used clinically, though drug resistance and side effects remain challenges [113]. Quinazolinone derivatives, showing promise as EGFR inhibitors with diverse biological activities, are under active investigation to enhance therapeutic specificity and efficacy against cancer.

Zayed et al. synthesized and evaluated a series of fluoro-quinazolinone derivatives for antitumor activity against MCF-7 and MDAMB-231 cells [114]. Among them, derivative (**125**) displayed strong inhibition of MCF-7 cells (IC_50_ = 12.44 ± 5.73 µM) and MDAMBA-231 cells (IC_50_ = 0.43 ± 0.02 µM), outperforming erlotinib (Figure 42). It also showed potent inhibition of EGFR (IC_50_ = 545.38 ± 0.04 nM) and microtubulin (IC_50_ = 6.24 µM). Molecular docking simulations identified key hydrogen bonds between derivate (**125**) and amino acids TyrA224, GlnA111, GlnB247, and LeuB248, contributing to a higher binding energy score (−24.7 kcal/mol) compared to colchicine (−11.1 kcal/mol). These docking results aligned with the compound’s experimental efficacy, elucidating its binding interactions with EGFR and microtubulin.

In a recent study, EI-Gazzar et al. synthesized novel 2-mercapto-quinazolin-4-one analogs and examined their in vitro anticancer activity, dihydrofolate reductase (DHFR) inhibition, and epidermal growth factor tyrosine kinase (EGFR-TK) pathway [115]. Notably, compound (**126**), featuring a 2-benzylthio moiety, exhibited broad-spectrum antitumor activity with high selectivity and safety (Figure 42). While it showed moderate EGFR-TK inhibition (IC_50_ = 13.40 nM), its DHFR inhibition potency was measured at 0.30 µM, somewhat less potent than methotrexate (IC_50_ = 0.08 µM). Compound (**126**) also induced cell cycle arrest and apoptosis in COLO-205 colon cancer cells. Molecular docking studies indicated a binding mode similar to gefitinib, with a *π*-interaction Lys745, suggesting promising potential as an anticancer agent.

Le et al. synthesized and evaluated new 3-methylquinazolinone derivatives for their in vitro antitumor effects against EGFRwt-TK and three human cancer cell lines: A549, PC-3, and SMMC-7721 [90]. Among these, quinazolin-4-one derivative (**127**) exhibited significant activity, with IC_50_ values of 12.30 ± 4.12 µM for A549, 17.08 ± 3.61 µM FOR PC-3, and 15.68 ± 1.64 µM for SMMC-7721 (Figure 42). Additionally, compound (**127**) induced late apoptosis in A549 cells at high concentrations and caused cell cycle arrest in the G2/M phase. For EGFRwt-TK inhibition, compound (**127**) showed an IC_50_ of 10 nM. Molecular docking studies revealed that its inhibitory activity likely results from stable hydrogen bonds formed with the residues R817 and T830 in EGFRwt-TK. Furthermore, interactions with the cationic residue K72 were suggested, highlighting a potential molecular interaction mechanism.

Walid M. Ghorab and his team developed and synthesized a series of quinazolinone derivatives based on 2-mercapto-3-phenylquinazolinones and assessed their cytotoxic effects on the HepG-2 hepatocellular carcinoma cell line [116]. Compounds (**128**) and (**129**) demonstrated IC_50_ values of 1.11 µM and 4.28 µM, respectively, surpassing the activity of the reference compound, adriamycin (IC_50_ = 32.02 µM) (Figure 42). The EGFR inhibitory effects of these compounds yielded IC_50_ values of 73.23 µM and 58.26 µM, respectively, which are higher than that of erlotinib (IC_50_ = 9.79 µM). Modeling studies indicated that compound (**128**) achieved higher docking scores within the EGFR active site than erlotinib. Furthermore, γ-irradiation enhanced the cytotoxic effects of compounds (**128**) and (**129**) on tumor cells, allowing for dose reduction and potentially lowering adverse effects.

Fang et al. synthesized a series of novel quinazolinone hydrazide derivatives as EGFR inhibitors [117]. Among these, compound (**130**) showed the most potent antitumor activity, with IC_50_ values of 1.31 µM for MCF-7, 1.89 µM for HepG2, and 2.10 µM for SGC (Figure 42). It also demonstrated high EGFR inhibitory activity, with an IC_50_ of 0.59 µM. Molecular docking studies revealed that compound (**130**) binds effectively to the ATP binding site of EGFR, interacting with VAL702, ASP831, LYS721, and MET769 within the active pocket through robust hydrogen bonds, along with Pi-Sigma and Pi-cation interactions. These results position quinazolinone hydrazide derivatives as promising anticancer agents targeting EGFR.

In a recent study, Kothayer et al. introduced a series of novel quinazolinone-based streptozotocin derivatives, designed and synthesized to triple-target double-mutant EGFR^L858R/T790M^, COX-2, and 15-LOX [118]. Compounds (**131**), (**132**), and (**133**) demonstrated low micromolar IC_50_ values against these targets, displaying selectivity for COX-2 over COX-1 and for EGFR^L858R/T790M^ over wild-type EGFR (Figure 43). Additionally, compounds (**132**) and (**133**) exhibited significantly higher nitric oxide (NO) production compared to celecoxib and indomethacin. Compounds (**131**) and (**133**) showed notable antiproliferative activity against the breast cancer cell line BT-459, with inhibition rates of 67.14% and 70.07%, respectively. Ligand–receptor binding studies further supported their strong binding affinity, highlighting their potential as effective multi-targeting agents.

Quinazolinone derivatives are integral to developing EGFR inhibitors for cancer therapy, with their efficacy shaped by specific structural features and SARs. The quinazolinone core structure, essential for EGFR binding, engages in hydrogen bonding through its 4(3H)-oxo group. The introduction of polar substituents, such as amino, hydroxyl, and amide groups at the 6-position, enhances binding affinity by increasing the hydrogen bonding potential. Hydrophobic substituents like halogens or phenyl groups at the 7-position improve interactions through hydrophobic effects and π-π stacking (Figure 44). Furthermore, aryl or heteroaryl groups at the 2-position strengthen binding affinity by reinforcing hydrophobic interactions. In conclusion, the exploration of quinazolinone derivatives for EGFR inhibition offers promising opportunities, particularly through structural optimization and strategic addition of specific substituents. Future research should focus not only on refining existing compounds but also on investigating novel quinazolinone derivatives to bolster their role in cancer treatment. Through continued research and technological advances, these compounds show significant potential effective agents in anticancer treatment.

### 3.7. PI3K Inhibitors

Phosphatidylinositol-3 kinases (PI3Ks) are a family of lipid kinases that phosphorylate the 3′-OH group of inositol phospholipids, classified into three classes based on their catalytic subunits, which range from 110 to 120 kDa in molecular weight [119]. Dysregulation of PI3K is implicated in various human malignancies [120], including breast [121], colon [122], endometrial [123], and prostate [124]. The PI3K kinase isoforms α, β, δ, and γ are encoded by the PIK3CA, PIK3CB, PIK3CD, and PIK3CG genes, respectively. Mutations or overexpression of these isoforms often contribute to treatment failure in cancer therapy. The PI3K pathway, crucial for cell growth, survival, and proliferation, is thus a promising target for cancer treatment [125]. Understanding PI3K activity in quinazolinones may reveal their potential as therapeutic agents for cancer and other PI3K-related diseases. Future studies should focus on the conformational relationship between quinazolinones and their PI3K inhibition mechanism to develop potent, selective inhibitors with enhanced efficacy.

Wani et al. synthesized a novel quinazolinone chalcone derivative, compound (**134**), which demonstrated significant anticancer properties in both in vitro and in vivo studies, effectively inhibiting cell proliferation across multiple cancer cell lines (Figure 45) [126]. Compound (**134**) induces apoptosis by enhancing V-FITC protein binding, increasing the G0 cell fraction, reducing mitochondrial membrane potential, lowering the Bcl-2/Bax ratio, and generating apoptotic vesicles. Additionally, it significantly impacts the PI3K/Akt/mTOR signaling pathway and regulates cell cycle proteins, including Skp-2, p21, and p27, facilitating the transition of HCT-116 cells into the S phase and G2/M phase. Liang et al. designed and synthesized quinazolinone derivative (**135**), which exhibited an IC_50_ of 13.11 nM against PI3Kγ kinase [127]. This compound showed cytotoxic effects on leukemia cells across 12 different tumor cell lines. Mechanistic studies revealed that (**135**) exerts antiproliferative activity by inhibiting PI3K-AKT signaling and activating phosphorylated p38 and ERK in both human and murine leukemia cells (Figure 45). This compound holds promise as a novel therapeutic agent for further exploration in cancer therapy.

Kim et al. synthesized a series of quinazolinone derivatives and evaluated their inhibitory effects on the PI3K enzyme, along with their anticancer activity in hematologic malignant cell lines [128]. Among these derivatives, compounds (**136**) (IC_50_ = 0.39 nM) and (**137**) (IC_50_ = 0.09 nM) exhibited notable enzymatic activity, with compound (**137**) showing approximately fourfold selectivity for PI3Kγ/δ compared to idelalisib, a PI3Kδ inhibitor. (Figure 45) The potency and selectivity toward PI3Kδ were attributed to modifications in the quinazolinone ring and the addition of a hydrophobic cyclopropyl group with fluorine or methyl substituents. Additionally, compounds (**136**) and (**137**) demonstrated targeted effects in cancer cells, promoting apoptosis. The antitumor efficacy of compound (**136**) was validated in xenograft models, where it effectively inhibited the PI3K pathway by reducing levels of p-AKT, p-S6, and p-4EBP1 in tumor tissues. These results suggest the potential of compound (**136**) as a therapeutic agent for treating hematologic malignancies.

Khalifa et al. synthesized 2-(pyridin-4-yl)quinazolin-4(3H)-ones as potential PI3K inhibitors and assessed their antiproliferative properties against HePG-2, MCF-7, and HCT116 cancer cell lines [129]. Among these, compound (**138**) displayed anti-HePG2 activity with an IC_50_ of 60.29 ± 1.06 µM, comparable to adriamycin (IC_50_ = 69.60 ± 1.50 µM). Compounds (**139**) and (**140**) showed superior inhibitory activity against HePG2, with IC_50_ values of 104.94 ± 2.46 µM and 126.40 ± 1.83 µM, respectively (Figure 46). The ADP-Glo assay confirmed the PI3K inhibitory potential, with compounds (**138**) and (**140**) exhibiting IC_50_ values of 31.92 ± 3.26 µM and 74.48 ± 2.91 µM, respectively, close to that of LY294002 (IC_50_ = 57.30 ± 2.02 µM). These compounds also interacted with Lys779 through additional hydrogen bonds. Kinase inhibition evaluation, along with docking studies, indicated favorable binding affinities, supporting the development of new anticancer candidates. Based on these findings, further research should focus on optimizing the structures of these compounds to enhance their efficacy as PI3K inhibitors and anticancer agents.

According to the molecular docking results, it was found that the binding sites of the quinazolinone structure with PI3K were mainly concentrated in the ATP-binding site of PI3Kα and its surrounding amino acid residues, such as PRO-810, ILE-963, ASP-964, and TYR-867 (Figure 47). The interaction of these binding sites helps to inhibit the activity of PI3K, thus exerting antitumor effects. However, it should be noted that there are multiple isoforms of PI3K with differences in the amino acid residues of different isoforms, so isoform selectivity needs to be considered in drug design to improve the efficacy and safety of drugs.

Quinazolinones are promising PI3K inhibitors for antitumor drug development, targeting PI3K a key regulator of cell growth, survival, and metabolism whose abnormal activation is linked to cancer. Quinazolinone derivatives interact with PI3Kγ/δ, inhibiting activity by disrupting substrate binding or inducing conformational changes. Additionally, these compounds can target the PI3K signaling pathway, including PI3K, AKT, and mTOR (mammalian target of rapamycin), and inhibit AKT phosphorylation, indirectly inhibiting AKT through PI3K suppression. Quinazolinone derivatives also inhibit mTOR in preclinical studies, potentially by competing for substrate binding or reducing expression. Despite certain challenges, quinazolinone derivatives hold promise as new PI3K inhibitors in cancer therapy. Future studies should aim to optimize structures, explore combination therapies, and improve clinical monitoring. Addressing these challenges could enhance the therapeutic potential of quinazolinone derivatives in cancer treatment, offering additional options for patients.

**Table 2 biomolecules-15-00210-t002:** Overview of potent quinazolinone-based antitumor agents.

Compound Number	Structure	Activity Tested Against the Cells	Cytotoxicity	Reference
**141**	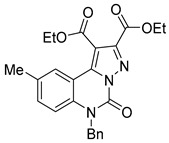	A549	IC_50_ = 14.2 μM	[130]
**142**	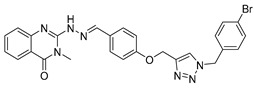	(i) EBC-1 (ii) A549 (iii) HT29 (iv) U-87MG	IC_50_ (μM) (i) 8.6 (ii) 64.9 (iii) 65.2 (iv) 24.6	[131]
**143**	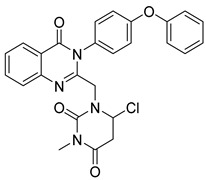	(i) MCF-7 (ii) SW480 (iii) MRC-5	IC_50_ (μM) (i) 21.5 (ii) 1.1 (iii) 105	[132]
**144**	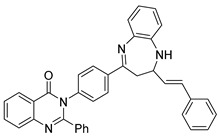	(i) MCF-7 (ii) HepG-2 (iii) HCT-116	IC_50_ (μM) (i) 19.2 (ii) 24.5 (iii) 14.2	[133]
**145**	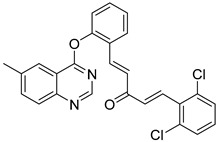	(i) MGC-803 (ii) PC-3 (iii) Bcap-37	IC_50_ (μM) (i) 0.85 (ii) 1.37 (iii) 4.98	[134]
**146**	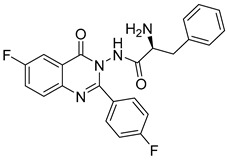	(i) MCF-7 (ii) MDA-MB-231	IC_50_ (μM) (i) 0.44 (ii) 24.67	[114]
**147**	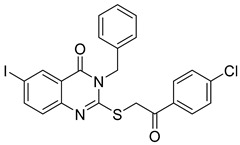	(i) MCF-7 (ii) HeLa (iii) HepG-2 (iv) HCT-8	IC_50_ (μM) (i) 3.76 (ii) 4.98 (iii) 4.17 (iv) 9.5	[135]
**148**	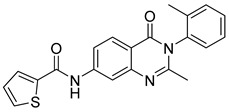	(i) RPMI-8226 (ii) K-562 (iii) HL-60	IC_50_ (μM) (i) 8.0 (ii) 12.8 (iii) 19.2	[136]
**149**	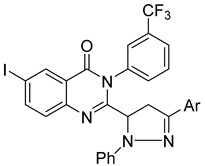	(i) HepG-2 (ii) MCF-7 (iii) A-549	IC_50_ (μM) (i) 9.08 (ii) 13.85 (iii) 108.08	[137]
**150**	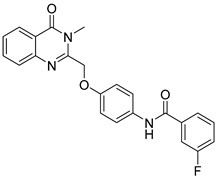	(i) A431 (ii) A549 (iii) MCF-7 (iv) NCl-H1975	IC_50_ (μM) (i) 3.48 (ii) 2.55 (iii) 0.87 (iv) 6.42	[138]
**151**	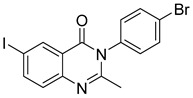	(i) HL-60 (ii) U937	IC_50_ (μM) (i) 22.1 (ii) 31.5	[139]
**152**	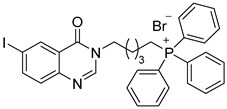	(i) HepG-2 (ii) A549 (iii) MCF-7 (iv) QSG-7701	IC_50_ (μM) (i) 14.52 (ii) 7.51 (iii) 6.56 (iv) 10.61	[140]
**153**	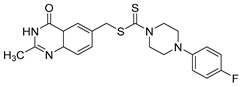	K562	IC_50_ = 0.5 μM	[141]
**154**	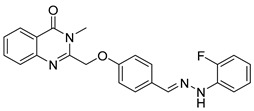	(i) PC-3 (ii) A549	IC_50_ (μM) (i) 7.73 (ii) 7.36	[142]
**155**	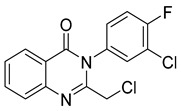	(i) SW1116 (ii) A549 (iii) MCF-7	IC_50_ (μM) (i) 9.5 (ii) 9.3 (iii) 5.8	[143]
**156**	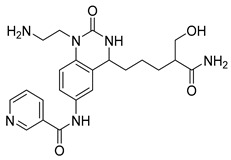	(i) HepG-2 (ii) A2780 (iii) MDA-MB-231	IC_50_ (μM) (i) 37.59 (ii) 22.76 (iii) 85.69	[144]
**157**	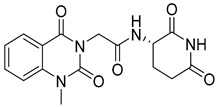	(i) HepG-2 (ii) PC-3 (iii) MCF-7 (iv) HCT-116	IC_50_ (μM) (i) 26.71 (ii) 22.11 (iii) 9.25 (iv) 16.09	[145]
**158**	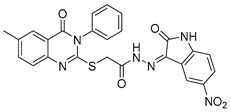	(i) LOVO (ii) MDA-MB-231	IC_50_ (μM) (i) 9.91 (ii) 10.38	[146]
**159**	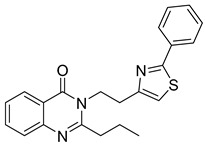	(i) HT-29 (ii) PC-3 (iii) MCF-7	IC_50_ (μM) (i) 12 (ii) 10 (iii) 10	[147]
**160**	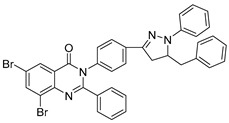	MCF-7	IC_50_ = 1.7 μM	[148]
**161**	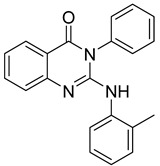	(i) MCF-7 (ii) HCT-116	IC_50_ (μM) (i) 14.7 (ii) 4.87	[149]
**162**	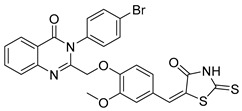	(i) HL-60 (ii) K-562	IC_50_ (μM) (i) 1.2 (ii) 1.5	[150]

## 4. Challenges and Future Directions

Despite significant advancements in the research and development of quinazolinone drugs, substantial efforts are still required to transform them into antitumor therapeutics effectively. First, while multiple synthetic methods for quinazolinones exist, their operational complexity, stringent reaction conditions, high production costs, and environmental impact pose significant challenges to large-scale production and practical application. Second, the exploration of synthetic pathways and the evaluation of product activity have mainly been conducted independently, leading to a lack of targeted development of the biological activities of synthesized compounds. The emphasis has predominantly been on methodological innovation rather than practical application. Third, current evidence supporting the antitumor efficacy of quinazolinones remains insufficient.

Therefore, future research on the antitumor activity of quinazolinones should prioritize the following areas. Firstly, the development of more efficient and environmentally friendly synthetic strategies is crucial. Emerging biosynthetic approaches utilizing microorganisms or enzymes offer promising alternatives due to their mild reaction conditions and high selectivity. Secondly, targeted synthesis of quinazolinone compounds with enhanced biological activity should be pursued, integrating synthetic methodologies with product activity assessment. Advances in computational chemistry and machine learning present valuable opportunities for optimizing drug design and synthesis. Finally, focusing on developing target-specific drugs that address key stages of tumor progression may represent a major direction for future antitumor drug development.

## 5. Conclusions

Quinazolinones are a promising class of compounds in anticancer drug development, with structural characteristics that support their potential to inhibit tumor growth. Recent research has extensively investigated the antitumor mechanisms of quinazolinones, aiming to clarify their actions within cancer cells and evaluate their potential clinical applications. While some mechanisms have been identified, further research is needed to gain a comprehensive understanding of quinazolines’ specific actions across various cancer types, thus improving the precision of pharmacological assessments and aiding in optimized drug design. As knowledge of quinazolinone antitumor activity progresses, future research should prioritize clinical trials to confirm their efficacy and safety in cancer patients. These trials will be critical to advancing the clinical application of quinazolinones.

Additionally, exploring combination therapies that integrate quinazolinones with other anticancer agents may enhance therapeutic outcomes. Investigating the synergistic effects of quinazolinones with current chemotherapy or immunotherapy agents represents another promising research direction. Quinazolinones continue to gain attention as potential antitumor agents, with their ability to inhibit cancer cell proliferation and metastasis through multiple mechanisms offering new insights into cancer treatment. As research advances, the prospective application of quinazolinones in cancer therapy appears increasingly promising, and further studies are anticipated to provide additional support and data to inform their clinical use.

## Figures and Tables

**Figure 1 biomolecules-15-00210-f001:**
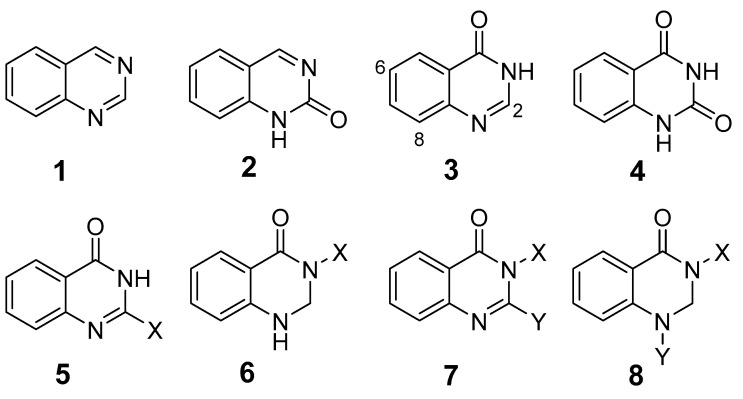
Quinazoline and quinazolinone structure.

**Figure 2 biomolecules-15-00210-f002:**
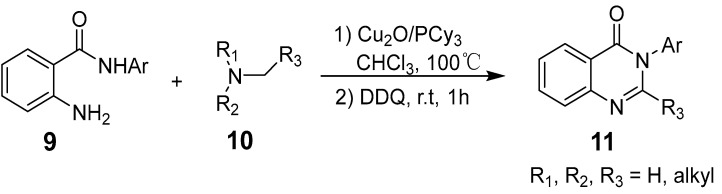
Copper-catalyzed synthesis of quinazolinones.

**Figure 3 biomolecules-15-00210-f003:**
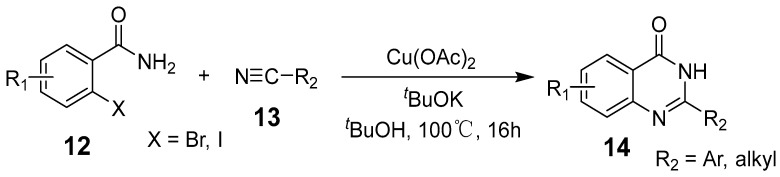
Copper-catalyzed synthesis of quinazolin-4(3H)-ones.

**Figure 4 biomolecules-15-00210-f004:**
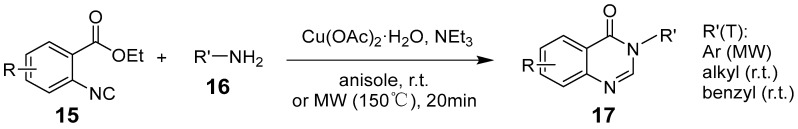
Copper-catalyzed synthesis of quinazolin-4-ones.

**Figure 5 biomolecules-15-00210-f005:**
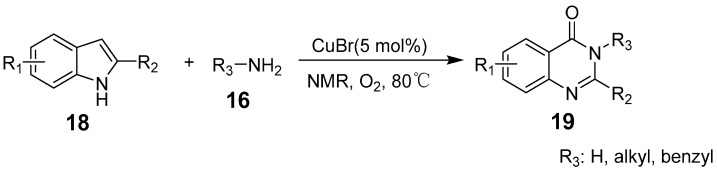
Copper-catalyzed synthesis of 2-arylquinazolinones.

**Figure 6 biomolecules-15-00210-f006:**
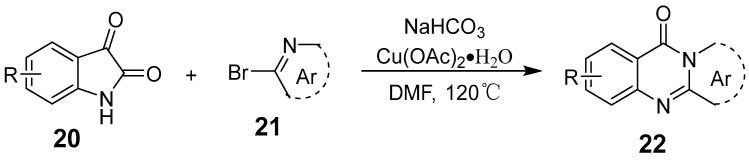
Copper-catalyzed synthesis of pyrido-fused quinazolinones.

**Figure 7 biomolecules-15-00210-f007:**
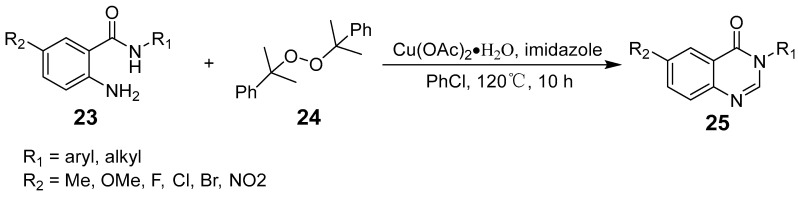
Copper-catalyzed cascade reactions for the synthesis of quinazolinones.

**Figure 8 biomolecules-15-00210-f008:**
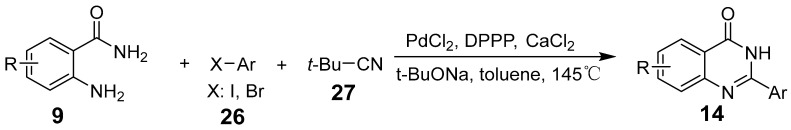
Palladium-catalyzed one-pot synthesis of quinazolin-4(3H)-ones.

**Figure 9 biomolecules-15-00210-f009:**
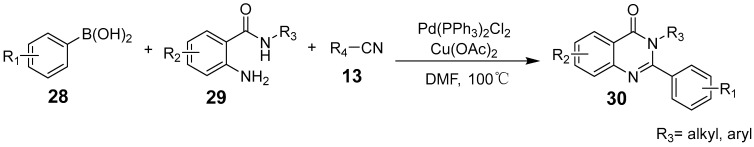
Palladium-catalyzed 2,3-disubstituted quinazolinones.

**Figure 10 biomolecules-15-00210-f010:**
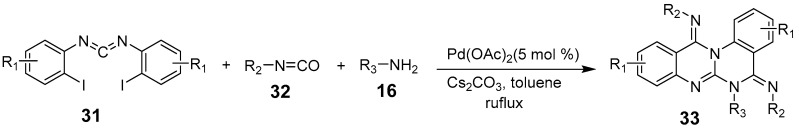
Palladium-catalyzed synthesis of quinazolino[3,2-a]quinazolines.

**Figure 11 biomolecules-15-00210-f011:**
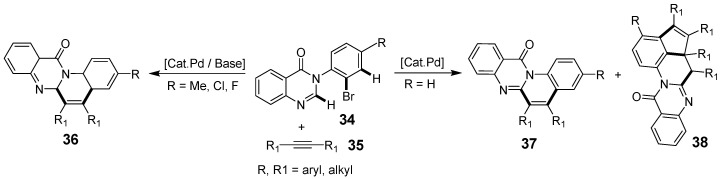
Palladium-catalyzed synthesis of quinazolin-4(3H)-ones with alkyne.

**Figure 12 biomolecules-15-00210-f012:**
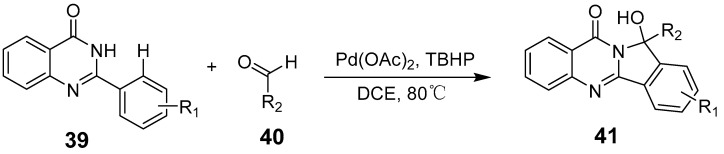
Palladium-catalyzed synthesis of hydroxyisoindolo[1,2-b]quinazolinones.

**Figure 13 biomolecules-15-00210-f013:**
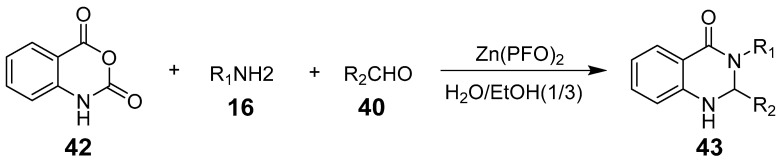
Zinc(II)-catalyzed synthesis of 2,3-disubstituted quinazolin-4(1H)-ones.

**Figure 14 biomolecules-15-00210-f014:**
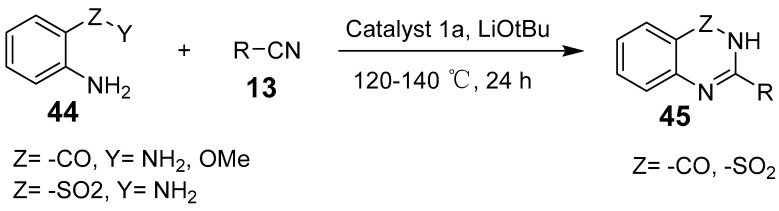
Zinc (II)-stabilized amidyl radical-promoted synthesis of quinazolinones.

**Figure 15 biomolecules-15-00210-f015:**
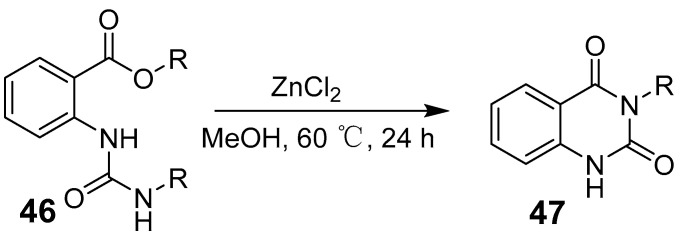
Cyclization of acyclic urea derivatives to quinazolinones.

**Figure 16 biomolecules-15-00210-f016:**
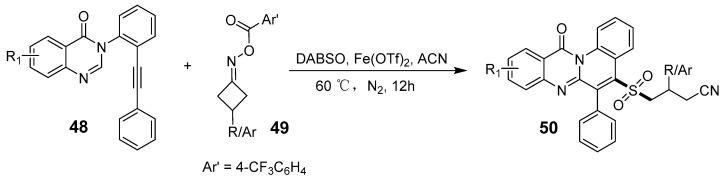
Iron-catalyzed cyclization to access cyanoalkyl sulfonyl quinolino-quinazolinones.

**Figure 17 biomolecules-15-00210-f017:**
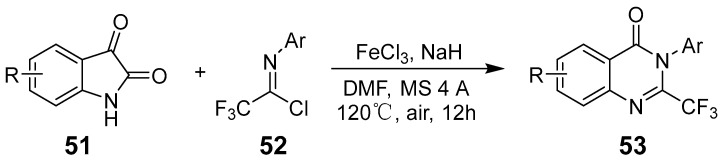
Iron (III)-mediated synthesis of 2-(trifluoromethyl)quinazolinones.

**Figure 18 biomolecules-15-00210-f018:**
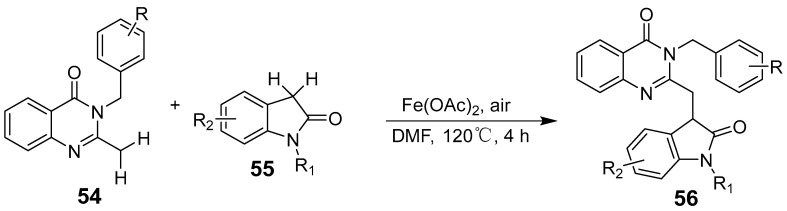
Iron (II)-catalyzed diastereoselective cross-dehydrogenative.

**Figure 19 biomolecules-15-00210-f019:**
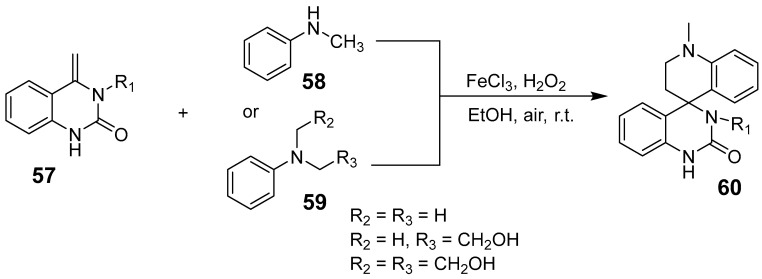
Iron-catalyzed synthesis of quinoline-spiro-quinazolinones.

**Figure 20 biomolecules-15-00210-f020:**
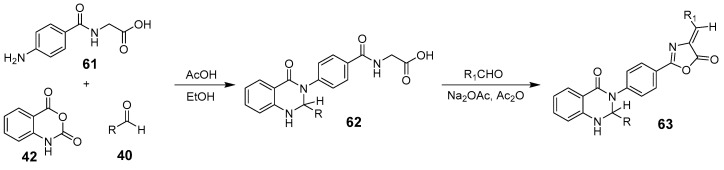
MW-assisted synthesis of 2,3-dihydroquinazolin-4(1H)-ones.

**Figure 21 biomolecules-15-00210-f021:**
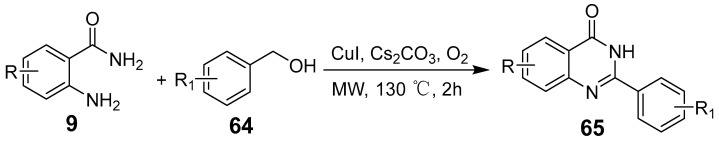
Microwave-assisted synthesis of quinazolin-4(3H)-ones.

**Figure 22 biomolecules-15-00210-f022:**
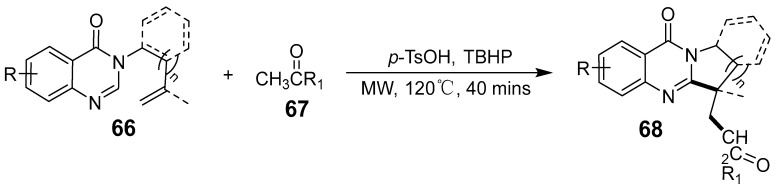
Synthesis of keto alkyl-substituted polycyclic quinazolinones under MW irradiation.

**Figure 23 biomolecules-15-00210-f023:**
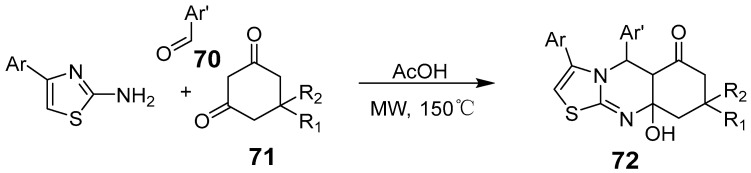
Synthesis of thiazolo[2,3-b] quinazolinones under MW irradiation.

**Figure 24 biomolecules-15-00210-f024:**
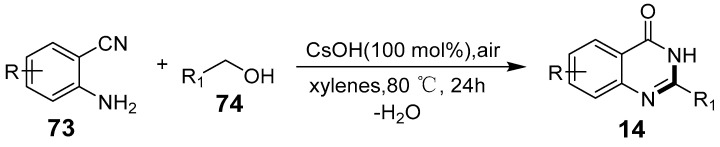
CsOH-mediated synthesis of quinazolin-4(3H)-ones.

**Figure 25 biomolecules-15-00210-f025:**
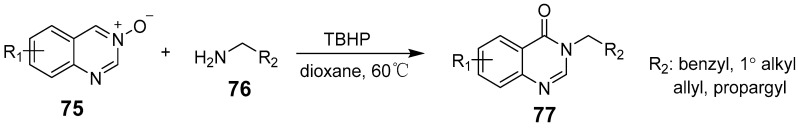
TBHP promoted the synthesis of quinazolinones.

**Figure 26 biomolecules-15-00210-f026:**
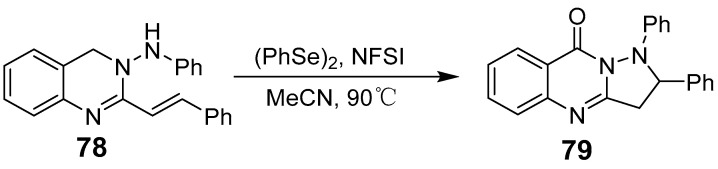
Synthesis of 1,2-diarylpyrazolo[5,1-b]quinazolin-9(1H)-ones.

**Figure 27 biomolecules-15-00210-f027:**
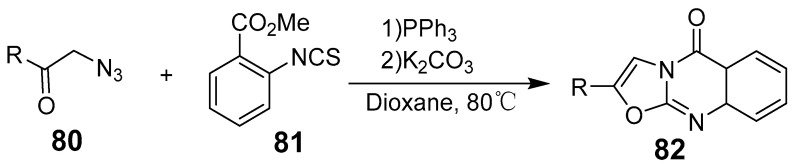
Synthesis of 2-substituted oxazolo-quinazolinones.

**Figure 28 biomolecules-15-00210-f028:**
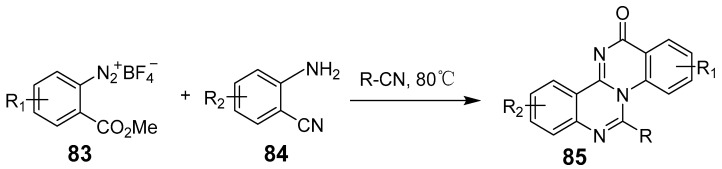
Preparation of quinazolino-quinazolinones via a cascade approach.

**Figure 29 biomolecules-15-00210-f029:**
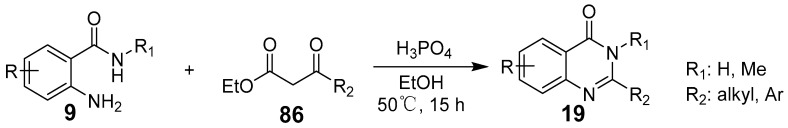
Phosphorous acid-catalyzed synthesis of 2-substituted quinazolin-4(3H)-ones.

**Figure 30 biomolecules-15-00210-f030:**
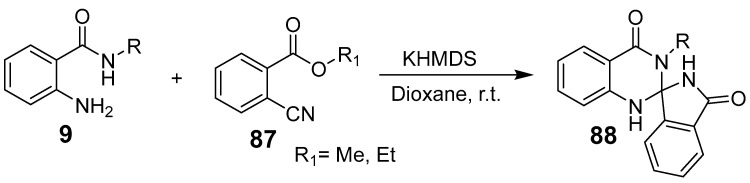
Synthesis of spiroiso indolinone dihydro-quinazolinones.

**Figure 31 biomolecules-15-00210-f031:**
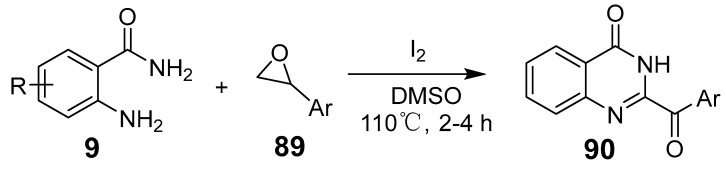
Iodine promoted the synthesis of aza heterocycles.

**Figure 32 biomolecules-15-00210-f032:**
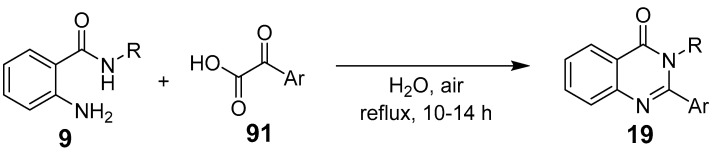
Synthesis of quinazolinones from *α*-Keto acids.

**Figure 33 biomolecules-15-00210-f033:**
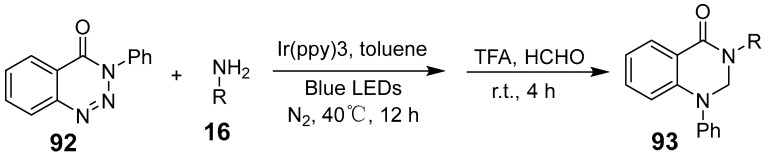
Visible-light photocatalyzed synthesis of quinazolinone.

**Figure 34 biomolecules-15-00210-f034:**
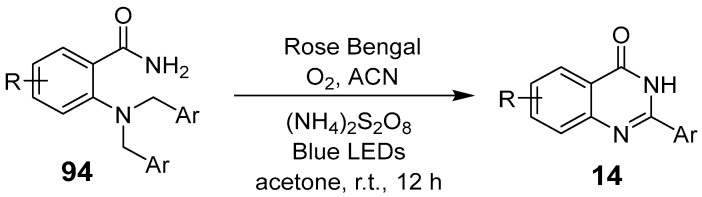
Photoredox-catalyzed nonaqueous oxidative C–N cleavage.

**Figure 35 biomolecules-15-00210-f035:**
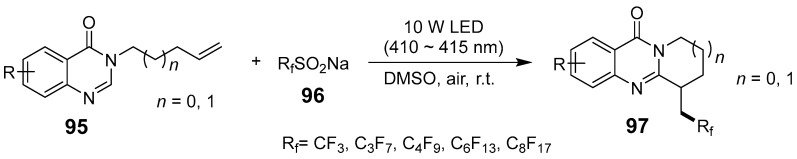
Self-catalyzed photo tandem synthesis of perfluoroalkyl-substituted quinazolinones.

**Figure 36 biomolecules-15-00210-f036:**
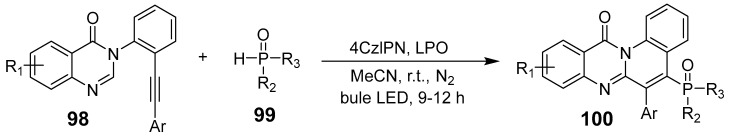
Synthesis of phosphoryl quinolino[2,1-b]quinazolinones.

**Figure 37 biomolecules-15-00210-f037:**
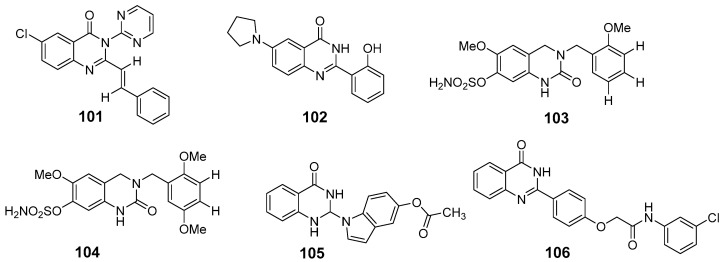
Quinazolinone-based derivatives as a tubulin polymerization inhibitor.

**Figure 38 biomolecules-15-00210-f038:**
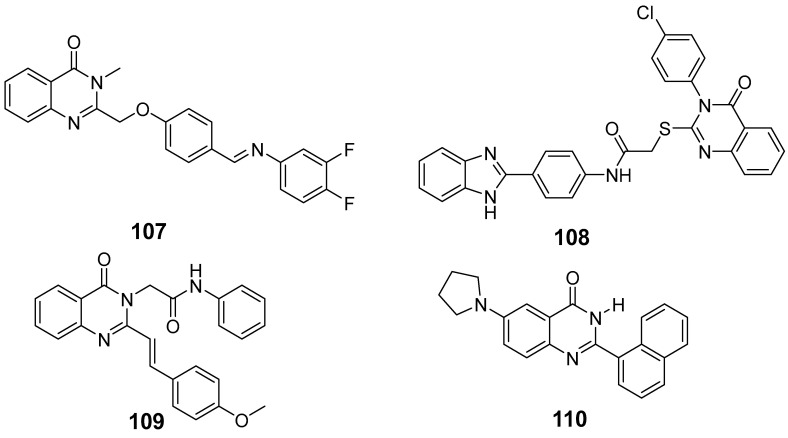
Quinazolinone-based derivatives induce cell cycle blockade.

**Figure 39 biomolecules-15-00210-f039:**
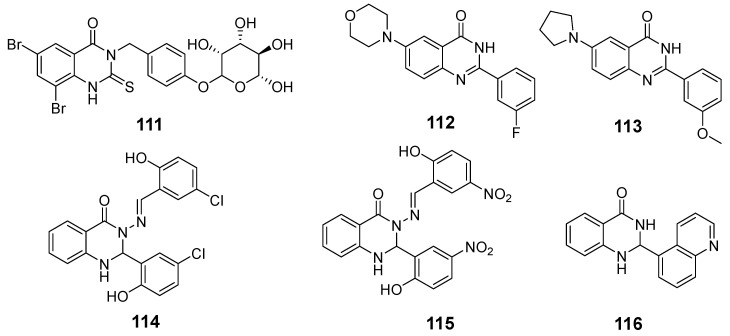
Quinazolinone-based derivatives induce apoptosis.

**Figure 40 biomolecules-15-00210-f040:**
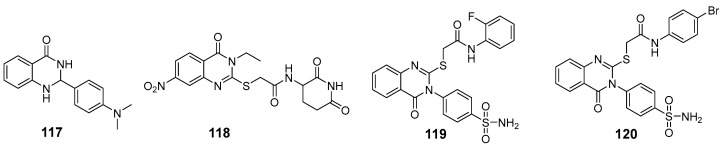
Quinazolinone derivatives with anti-inflammatory capacity.

**Figure 41 biomolecules-15-00210-f041:**
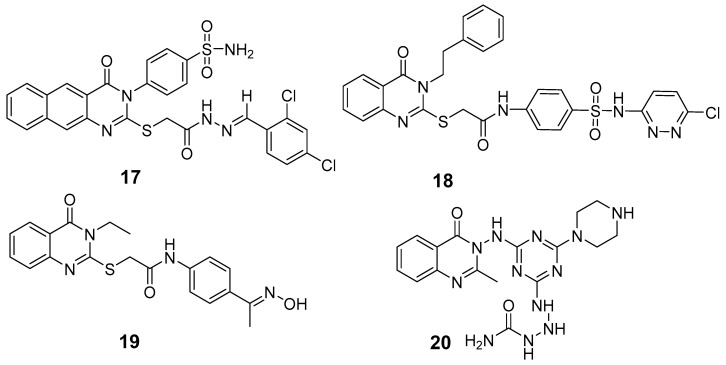
Quinazolin-4-one derivatives inhibit angiogenesis.

**Figure 42 biomolecules-15-00210-f042:**
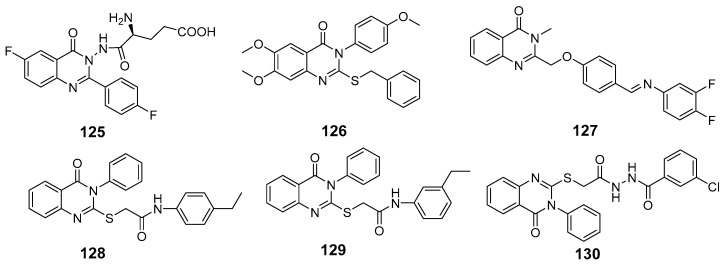
Quinazolinone-based derivatives as EGFR inhibitors.

**Figure 43 biomolecules-15-00210-f043:**
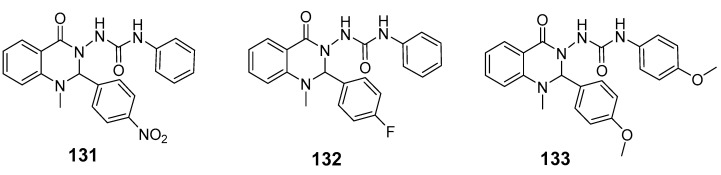
Quinazolinone-based streptozotocin derivatives as EGFR inhibitors.

**Figure 44 biomolecules-15-00210-f044:**
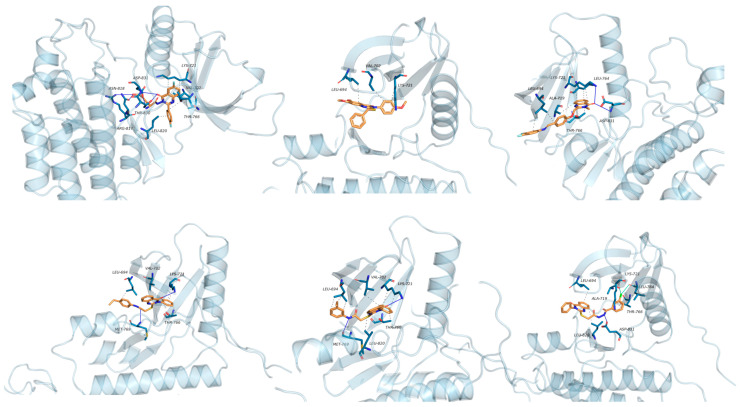
Interaction maps of derivatives 125–130 inside the active site of EGFR (PDB: 1M17).

**Figure 45 biomolecules-15-00210-f045:**
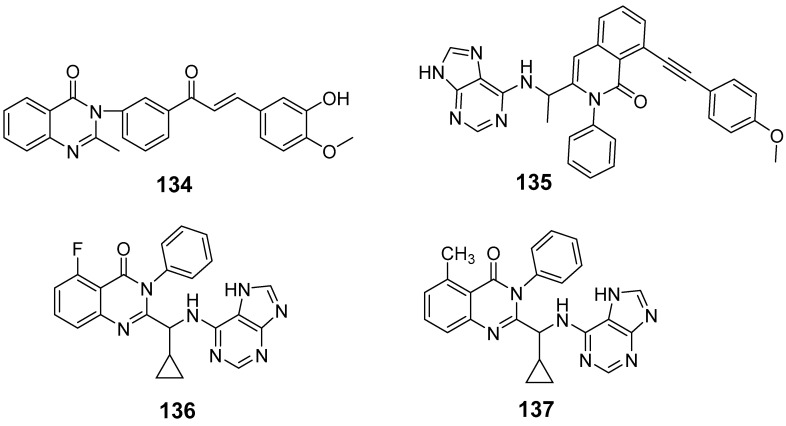
Quinazolinone derivatives as PI3K inhibitors.

**Figure 46 biomolecules-15-00210-f046:**
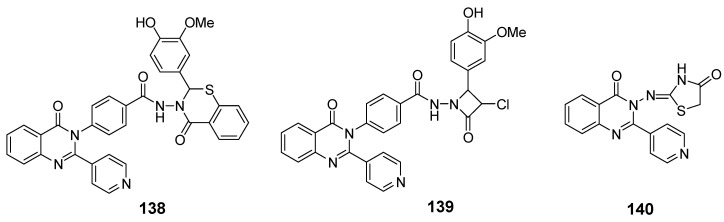
2-(Pyridin-4-yl)quinazolin-4(3H)-ones as PI3K inhibitors.

**Figure 47 biomolecules-15-00210-f047:**
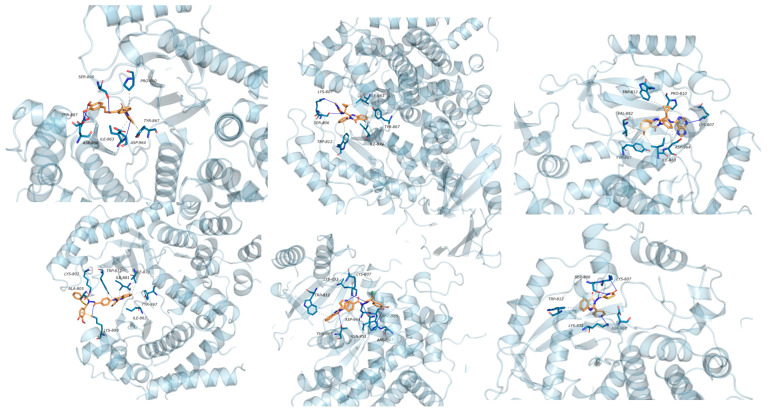
Interaction maps of derivatives 134–139 inside the active site of PI3K (PDB: 1E7U).

**Table 1 biomolecules-15-00210-t001:** Some marketed drugs available, which contain the quinazolinone moiety.

S.No	Drug	IUPAC Name	Activity	References
1	Afloqualone	6-amino-2-(fluoomethyl)3-(2-methylphenyl) quinazolin-4-one	Sedative, Hypnotic	[15]
2	Albaconazole	7-Chloro-3-[(2R,3R)-3-(2,4-difluorophenyl)-3-hydroxy-4-(1,2,4-triazol-1-yl)butan-2-yl]quinazolin-4-one	Antifungal	[16]
3	Cloroqualone	3-(2,6-Dichlorophenyl)-2-ethyl-4-quinazolinone	Sedative, Antitussive	[17]
4	Fluproquazone	4-(4-fluorophenyl)-7-methyl-1-propan-2-ylquinazolin-2-one	NSAID	[18]
5	Febrifugine	3-(3-((2R,3S)-3-hydroxypiperidin-2-yl)-2-oxopropyl) quinazolin-4-one	Antimalarial	[19]
6	Fenquizone	7-chloro-4-oxo-2-phenyl-2,3-dihydro-1*H*-quinazoline-6-sulfonamide	Diuretic	[20]
7	Halofuginone	7-Bromo-6chloro-3-[3-[(2S,3R)-3-hydroxy-2-piperidinyl]-2-oxopropyl]-4-quinazolinone	Coccidiostat, Anticancer	[21]
8	Ispinesib	*N*-(3-aminopropyl)-*N*-[(1R)-1-(3-benzyl-7-chloro-4-oxoquinazolin-2-yl)-2-methylpropyl]-4-methylbenzamide	Anticancer	[22]
9	Idelalisib	5-fluro-3-phenyl-2-[(1S)-1-(7H-purin-6-ylamino)propyl]quinazolin-4-one	Antihematological cancer	[23]
10	Isaindingotone	3-(4-hydroxy-3,5-dimethoxybenzyl)-2,3-dihydropyrrolo[2,1-b] quinazolin-9-one	Anti-inflammatory	[24]
11	Nolatrexed	2-Amino-6-methyl-5-(4-pyridylthio)-1H-quinazolin-4-one	Thymidylate synthase inhibitor	[25]
12	Piriqualone	3-(2-methylphenyl)-2-[(*E*)-2-pyridin-2-ylethenyl]quinazolin-4-one	Anticonvulsant	[26]
13	Sclerotigenin	6,7-dihydroquinazolino[3,2-a][1,4]benzodiazepine-5,13-dione	Anti-insectant	[27]
14	Raltitrexed	(2S)-2-[[5-[methyl-[(2-methyl-4-oxo-3H-quinazolin6-yl)methyl]amino]thiophene-2-carbonyl]amino]pentanedioic acid	Anticancer	[28]
15	Elinogrel	1-(5-chlorothiophen-2-yl) sulfonyl-3-[4-[6-fluoro-7-(methylamino)-2,4-dioxo-1*H*-quinazolin-3-yl]phenyl]urea	Antithrombosis	[29]
16	Tiacrilast	(*E*)-3-(6-methylsulfanyl-4-oxoquinazolin-3-yl) prop-2-enoic acid	Antiallergic	[30]
17	Rutaecarpine	8,8a,13,13a-tetrahydroindolo [2′,3′:3,4] pyrido[2,1-b] quinazolin-5(7H)-one	Alzheimer’s disease, Protective alkaloid	[31,32]
18	Quinethazone	7-chloro-2-ethyl-4-oxo-2,3-dihydro-1H-quinazoline-6-sulfonamide	Antihypertensive	[33]

## Data Availability

No new data were created or analyzed in this study.

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
