# Peer review of "Quinazolinones as Potential Anticancer Agents: Synthesis and Action Mechanisms"

_biomolecules, 2025, doi:10.3390/biom15020210_

Round 1

Reviewer 1 Report

Comments and Suggestions for Authors

The review article has been submitted to Biomolecules which focuses on synthesizing quinazolinone derivatives and investigation of their biological activity. The review has some major issues as below;
-First of all, there are many grammatical and spelling errors.  The
review should be checked throughout.

- Why the authors did not perform some docking studies on the quinazolinones

- Abstract section should be improved.

Comments on the Quality of English Language

The English could be improved to more clearly express the research.

Author Response

First of all, there are many grammatical and spelling errors.  The review should be checked throughout.

We tried our best to improve the manuscript and made some changes to the manuscript. These changes will not influence the content and framework of the paper. And here we did not list the changes but marked in yellow in the revised paper. We appreciate for Reviewers’ warm work earnestly and hope that the correction will meet with approval.

Why the authors did not perform some docking studies on the quinazolinones

For this question, we would like to explain that the docking study of quinazolinone compounds has not been done before because some of the compounds involved in the article have been molecular docked by the original author, and the discussion of molecular docking is not very important to the significance of the article. However, according to the suggestion of Reviewer 1, we performed molecular docking of the target parts (EGFR, PI3K) involved in the article (Figure 44, Figure 47), and discussed the difference of the force between the compound and the target according to the molecular docking results. The changes added to the article are marked with a yellow background.

 Abstract section should be improved.

Thanks for your reminder. According to the reviewer's suggestion, we have rewritten the abstract part, and the rewritten part has been marked with yellow background.

Reviewer 2 Report

Comments and Suggestions for Authors

The authors emphasized that quinazolinones, a derivative of quinazoline, exhibit diverse biological activities with applications in pharmaceuticals and insecticides. Highlighting that some derivatives have already been developed as commercial drugs, they drew attention to the critical need for new anticancer agents in response to the rising incidence of cancer. This article provides a comprehensive examination of the synthesis methods and antitumor mechanisms of quinazolinones, underlining their promising potential in this field. The authors aimed to highlight the value of further research into quinazolinone-based anticancer therapies, encouraging efforts in antitumor drug development and offering a resource to guide future research directions. However,

1. In the paragraph located in lines 25 and 41, some numbers from 1 to 8 are written in bold format. If you include these in parentheses, it will reduce the visually distracting appearance.

2. Please provide references by leaving a space between the last word of the sentence and the reference. All references need to be corrected accordingly.

3. Please provide the citation for Table 1 within the text before the table itself.

4. Please include the names of the metal elements involved in the reactions performed for quinazolinone synthesis, followed by their symbols in parentheses.

5. The citation for Figure 16 in line 182 should be formatted in bold text. Additionally, please check the citation of other figures in lines 343, 350, 370, 384, 385, and 396.

6. In the section titled '3 The Antitumor Mechanism of Quinazolinones' the compounds are numbered as 101, 102, etc. Could you also write these numbers within parentheses?

Author Response

  1. In the paragraph located in lines 25 and 41, some numbers from 1 to 8 are written in bold format. If you include these in parentheses, it will reduce the visually distracting appearance.

    Thank you for their valuable comments on the details of the article. According to the suggestions , the numbers 1 to 8 from line 25 to line 41 have been displayed in bold text and placed within parentheses. The changes have been highlighted with a yellow background.

  2. Please provide references by leaving a space between the last word of the sentence and the reference. All references need to be corrected accordingly.

    Suggests that there should be a space between each reference and the last word of the sentence. This problem has been corrected for the entire text. All Spaces have been underlined in red.

  3. Please provide the citation for Table 1 within the text before the table itself.

    We were really sorry for our careless mistakes. Thank you for your reminder. In response to the reviewer's suggestion to reference Table 1, the reference to Table 1 has been made on line 41. The changes have been highlighted with a yellow background.

  4. Please include the names of the metal elements involved in the reactions performed for quinazolinone synthesis, followed by their symbols in parentheses.

    In accordance with the reviewer's comments, the metal elements involved in the reaction have been added using parentheses after subsection "2.1. Metal-catalysis Reaction". The changes have been highlighted with a yellow background.

  5. The citation for Figure 16 in line 182 should be formatted in bold text. Additionally, please check the citation of other figures in lines 343, 350, 370, 384, 385, and 396.

    Thanks for your careful review. According to the reviewer's comments, the citations in lines 182, 343, 350, 370, 384, 385 and 396 of the article have been checked again, and the citations in each line have been displayed in bold text. The changes have been highlighted with a yellow background.

  6. In the section titled '3 The Antitumor Mechanism of Quinazolinones' the compounds are numbered as 101, 102, etc. Could you also write these numbers within parentheses?

    Thanks for your careful checks. We are sorry for our carelessness. The number of compounds used has been bracketed in the section titled'3 The Antitumor Mechanism of Quinazolinones' as recommended. The changes have been highlighted with a yellow background.

Reviewer 3 Report

Comments and Suggestions for Authors

This review focuses on the synthesis and action mechanisms of Quinazolinones. Overall, it is well-written, and the figures are well-present. It should be a good reference for some chemists and biologists who are interested in Quinazolinones.

  Some minor comments:

1.      Line 831, I guess that you mean “gain” not “again”. Please correct this.

Author Response

Line 831, I guess that you mean “gain” not “again”. Please correct this.

Thanks to the reviewer for reviewing the details of the article. Based on the reviewer's comments, the word "gain" has been corrected to "gain" in line 831 of the article. The changes have been highlighted with a yellow background.

Round 2

Reviewer 2 Report

Comments and Suggestions for Authors

The authors improved the manuscript based on the previous comments. It is now can be accepted.